# A 6-Locking Cycles All-Digital Duty Cycle Corrector with Synchronous Input Clock

**Shao-Ku Kao** 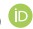

Department of Electrical Engineering and Green Technology Research Center, School of Electrical and Computer Engineering, College of Engineering, Chang Gung University, Taoyuan City 330, Taiwan; kaosk@mail.cgu.edu.tw

**Abstract:** This paper proposes an all-digital duty cycle corrector with synchronous fast locking, and adopts a new quantization method to effectively produce a phase of 180 degrees or half delay of the input clock. By taking two adjacent rising edges input to two delay lines, the total delay time of the delay line is twice the other delay line. This circuit uses a 0.18 μm CMOS process, and the overall chip area is 0.0613 mm$^2$, while the input clock frequency is 500 MHz to 1000 MHz, and the acceptable input clock duty cycle range is 20% to 80%. Measurement results show that the output clock duty cycle is 50% ± 2.5% at a supply voltage of 1.8 V operating at 1000 MHz, the power consumed is 10.1 mW, with peak-to-peak jitter of 9.89 ps.

**Keywords:** all digital; synchronization; fast locked; phase error; DCC; duty cycle





## 1. Introduction

In recent years, the development of CMOS process technology has driven the increased speed requirements for System-on-Chip (SoC), with clock signals widely used in most digital and mixed-signal circuits. To control circuit triggering under high-frequency operation, the accuracy of the frequency, phase, and pulse width of the clock signal is important for circuit applications. However, the Phase Locked Loop (PLL) [1–3] or Delay Locked Loop (DLL) [4–6], can only correct the signal frequency and phase, so the Duty Cycle Corrector (DCC) [7–20] was developed to correct the output duty cycle. The duty cycle of a clock plays a very important role in many circuit operations. If the duty cycle of the clock is distorted, the system may not work properly, thus raising the need for the duty cycle correction circuit.

In high-speed circuit design, to drive extremely large capacitive loads, a long series of inverter chains is often used to form a clock buffer. However, it is difficult to maintain the duty cycle of the final output signal of this clock buffer at 50%. The mismatch of the PMOS and NMOS transistors inside the inverter is also a factor to alter duty cycle of input clock. This problem becomes more serious in circuits with higher speeds and clock buffers with more stages. As a result, the clock duty cycle deviates significantly from 50%, and can become too wide or too narrow to thus cause the clock to disappear. In an Analog-to-Digital Converter (ADC) and Double-Data-Rate (DDR) SDRAM applications, both the positive and negative edges of the clock must capture data, thus the duty cycle must be close to 50%.

This paper proposes a new architecture to generate a half-cycle of the input clock. This method mainly proposes using two adjacent rising edges of the input clock, respectively, input to two delay lines with different delay cell time. After passing the number of delay cells, the rising edge of the two adjacent is synchronized, then the 180-degree phase and digital code of the input clock can be generated. Unlike the traditional TDC method, which generates the entire cycle of the digital code, this approach can effectively solve the parity counting problem resulting from the selection of the half-cycle digital code. The proposed circuit architecture can achieve synchronization with the input signal and 50% duty cycle in 6 cycles. The rest of the paper is organized as follows, the state-of-the-art are reviewed in Section 2. Section 3, explains the circuit architecture design. Section 4 introduces the

proposed DCC scheme. Section 5 provides measurement result and discussions, followed by conclusions in Section 6.

## 2. State-of-the-Art

Generally speaking, analog architectures of DCC [7–11] usually take a feedback form, which can have higher resolution and operating frequency, but with a lengthy lock time. Digital architectures are usually a non-feedback type [10–22], and thus have a faster lock time. The high degree of circuit integration and the easy design of Digital architectures make the realization of this circuit more advantageous and developmental than the analog architectures. However, digital architectures have low resolution and limited operating frequency cannot be compared with those of analog architectures.

The analog DCC circuits store the duty cycle in the form of the voltage by the continuous-time integrator or charge pump. This type of DCC has a lower correctable range of the input duty cycle and is highly sensitive to noise process, voltage and temperature (PVT) variations. The DCC proposed in [12] uses an adaptive charge pump generator with small current at low frequency and a larger current at high frequency. The control stage is formed by the current-starved pulse width with low power and is area efficient. The chopping technique is introduced in [13] to avoid the DC-offset problem in the charge pump. The Chopping technique improves the loop gain and is less sensitive to the PVT corners. Due to the chopping technique, the symmetrical structure, including the differential charge pump and differential chopper is included in [13].

The All-Digital Duty Cycle Corrector (ADDCC) does not contain any passive components, making it more convenient for system integration and use in low-voltage systems. The Time-to-Digital Converter (TDC) uses a quantized input signal to be converted into a digital code in a single cycle. The TDC suffers from some accuracy issues. For example, the sampling circuit (using D-type flip-flop, DFF) cannot distinguish between two edge of clock and D input signals because of limitations of setup time. For the problem arising when odd or even numbers when the digital code is divided by two, odd numbers will lose a 0.5 delay unit error, which will affect the accuracy of the 50% cycle signal. The ADDCC in Reference [17] uses a delay-recycle architecture to reduce the required delay line length to half of the reference clock signal (delay-recycled half-cycle time delay line). In addition to reducing the chip area, it can also work at a lower input frequency to achieve a wide operating frequency range at low cost. In Reference [18], a 1-bit digital duty-cycle detector is proposed to achieve a wide operating range and faster correction time in 14 clock cycles. The TDC structure is implemented in a 1-bit digital duty-cycle detector to detect the duty cycle ratio and convert to digital-bit. The time-to-digital converter in [18] achieves the input duty cycle of ADDCC of 10%~90%, but with low range, volume and working temperature. Directly finding the phase of a half cycle, instead of indirectly quantizing the entire cycle to generate a half phase, eliminates the problem of odd and even numbers, and the accuracy of the 50% duty cycle can be effectively improved. In Reference [18], a delay-locked loop to align the phase skew between input and output clock, and achieve low sensitivity to PVT variation for DRAM applications is proposed. An open-loop DCC in [21] uses an advanced digital-falling edge modulator. The digital-falling edge modulator modulates each falling edge of input clock and inverted input clock, and stretches or shrinks by as much as half of the edge difference. Afterwards, the phase interpolator is performed and to interpolate the two modulated clock, the final output clock generated with 50% duty cycle in [22] has benefits with respect to covering a wide frequency range and fast locking time in 5 cycles. The relaxation oscillator is designed in [22] to detect duty-cycle and quadrature phase errors by converting them into two frequencies. Those two frequencies are compared and digitized. The digital frequency detector is implemented to compare those two frequencies and convert them to digital form. It achieves a wide range of duty cycle and quadrature phase errors. However, the relaxation oscillator suffers from nonlinearity and PVT variation can affect the correction performance and jitter. Therefore, the symmetrical and close placement of circuits is needed.

The traditional digital DCC only generates 50% of the output signal, and is not synchronized with the input clock. DCC combined with a delay-locked loop (DLL) is conventionally used to achieve input signal synchronization. Combining synchronization and a 50% output duty cycle creates design complications. The ADDCC of Reference [19] is mainly used to improve the problem of the half-cycle delay line mismatch, using a ring-type TDC for quantization to achieve fast locking, and uses three half delay lines to solve the delay line mismatch problem. Reference [20] is mainly divided into two parts: an all-digital delay-locked loop (ADDLL) and an all-digital duty cycle correction (ADDCC). The ADDLL mainly uses the TDC method to achieve fast locking. However, the half-cycle signal will have the problem of odd and even numbers and produce a large phase error. To solve this problem, a Weighted Signal Generator architecture is proposed to reduce the phase error. The traditional quantization method uses fine-tuning after TDC quantization to improve the error of the 50% signal. Even after fine-tuning, a 50% duty cycle signal will always have an error of half the delay cell time. If the problem of odd and even numbers can be effectively solved, the accuracy of the signal can be improved. The traditional solution is to compensate for the error of half the delay cell time.

### 3. Circuit Architecture Design

The digital duty cycle correction circuit needs to generate a phase of 180 degrees. We propose a novel method that uses two adjacent rising edges of input clock to be input to two delay lines with different delay times as follows:

From Figure 1, the first rising edge of the input signal starts to pass through the delay line to generate CK_A, and the second rising edge also passes through the delay line to generate CK_B, but the delay time passed by the first rising edge is $3\tau$ while the delay time passed by the second rising edge is $\tau$. When CK_A and CK_B reach synchronization, they can be represented by Equation (1), and a phase of 180 degrees can be obtained from Equation (1).

$$
\begin{aligned}
N \times 3\tau &= T + N \times \tau \\
N \times 2\tau &= T \\
N \times \tau &= \tfrac{T}{2}
\end{aligned}
\tag{1}
$$

where N is the number of delay units, $\tau$ is the delay time of the delay unit, and T is the cycle time of the input clock. According to the result of Equation (1), the number of delay cells required to delay half the input clock cycle is N as follows:

$$
\text{Half cycle}: \frac{T}{2} = N\tau
\tag{2}
$$

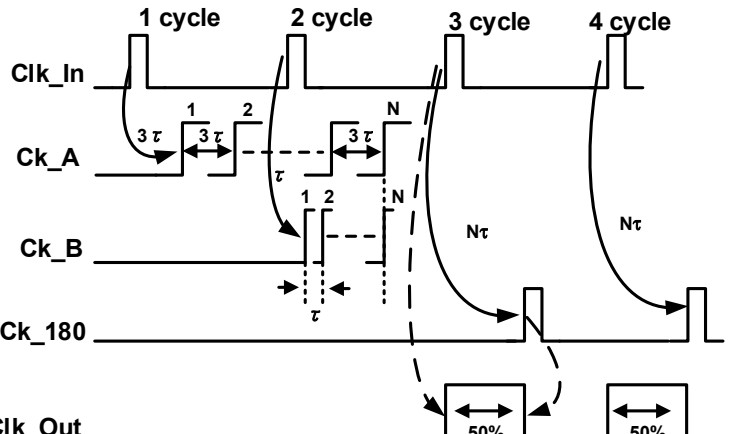

**Figure 1.** Concept diagram of proposed circuit.

The number of delay cells that delay the entire input clock cycle is 2N as shown below:

$$\text{Full cycle}: \text{T} = (2\text{N})\tau \tag{3}$$

From Equation (2), it follows that after the delay time of N delay units $\tau$, a phase difference of 180 degrees (Ck_180) can be obtained, as shown in Figure 1. To generate an output signal of 50% duty cycle, only the input signal for the rising edge of Clk_In and the rising edge of the signal Ck_180 are synthesized to produce the output signal Clk_Out. This method effectively solves the problem of odd and even numbers during full-digit quantization. At the same time, it can provide advantages including synchronization with the input signal, a 50% duty cycle clock and fast locking.

Figure 2 shows the circuit diagram for the entire architecture. For generation and input signal synchronization with a 50% duty cycle output signal requires simultaneous completion with DLL and DCC. Thus, we propose using a single architecture to complete synchronization with the input signal and generate a 50% duty cycle. This architecture is composed of four Half Delay Line (HDL), a Coarse Code Generator (CCG), a Fine Detector (FD), a Pulse Generator (PG), two Fine Delay Line (FDL) and an SR latch. The output digital code of the coarse adjustment code generator is used to achieve fast locking, and the fine adjustment detector is added to fine-tune the phase error to improve the resolution. Figure 3 shows the timing diagram of the overall architecture. The first and second cycles of Clk_In are mainly used to generate the digital codes H_code and F_code. When the rising edge of the first cycle of Clk_In is input to the pulse generator to generate the step signal P_In1, P_In1 passes through 3 HDLs (HDL1, HDL2, HDL3) to generate a multi-phase P1 signal. The second cycle rising edge of Clk_In is then input to the pulse generator to generate a step signal P_In2, which then passes through 1 HDL (HDL4) to generate a multi-phase P2 signal. When the multi-phase P1 signal and the multi-phase P2 signal are input to the coarse code generator circuit. The two digital codes, H_code and F_code, are generated when one of P1 and P2 are in-phase.

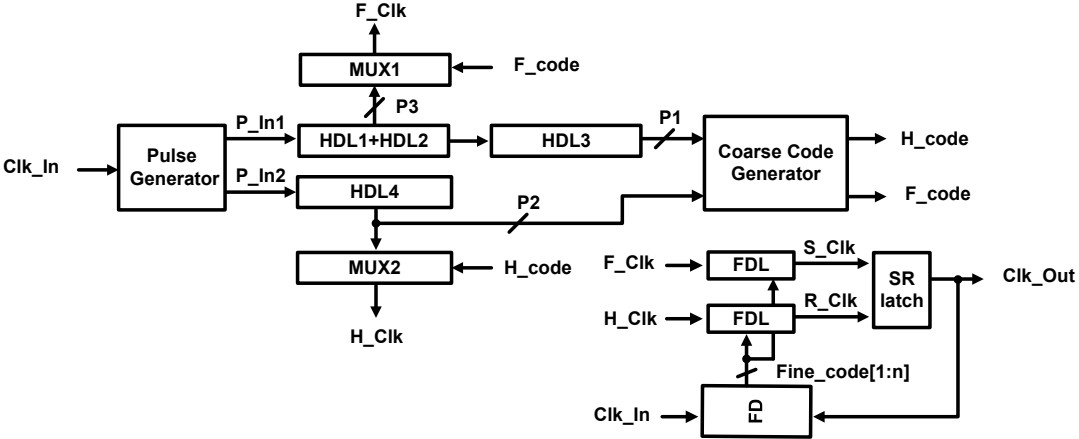

**Figure 2.** Block diagram of proposed circuit.

In the third cycle of Clk_In, H_code and F_code are used to control the delay lines. H_code controls the delay line HDL4 to generate the F_Clk signal, and F_code controls the delay lines HDL1 and HDL2 to generate the H_Clk signal. The F_Clk and H_Clk signals are synthesized to generate Clk_Out, which reaches 50% of the duty cycle. At this time, there is still a large phase difference T_Fine, as shown in Figure 3. The T_Fine represents the phase error of the Clk_In and Clk_Out signals. Therefore, the Clk_Out signal passes through the FD circuit to produce fine adjustment and generate the digital code Fine_Code [1: $n$] to control the FDL, so that the Clk_In and Clk_Out signal have a smaller phase error. This architecture requires a total of 6 cycles to achieve locking. The first three cycles are the coarse adjustment phase, while the fourth and fifth cycles of Clk_In are the fine adjustment phase, and the last cycle is the lock. The upper frequency of the proposed architecture is

limited by t1, which is the delay time of the pulse generator. The delay time of t1 must be less than one cycle. The lowest frequency depends on the number of delay units. If the operating frequency is higher, more delay units are needed.

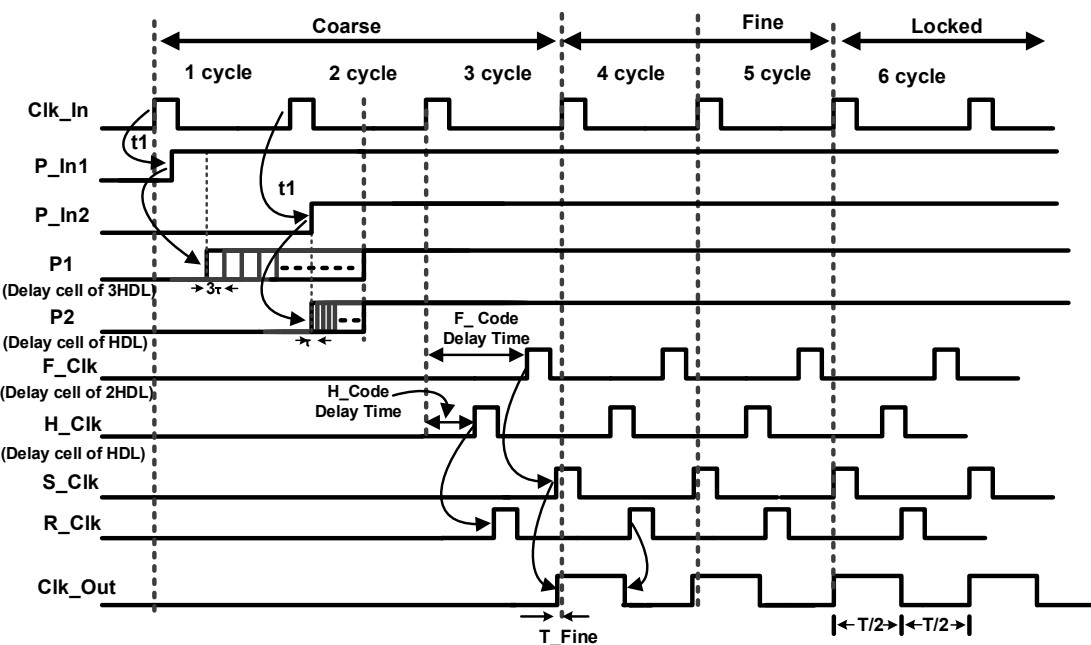

**Figure 3.** Time diagram of proposed circuit.

## 4. Circuit Architecture

### 4.1. Half-Delay Line

Figure 4a shows the proposed half-delay line, formed by using two inverters in series, used as the phases P1 to Pn required for coarse quantization. When it is used as the signal F_Clk in Figure 2, the signal to be used for H_Clk is P_Out. The P_Out signal is a delay cell using two inverters connected in series, and one of the inverters is added to the selected signals S1 to Sn, as shown in Figure 4a. The purpose is to improve the accuracy of the output signal Clk_Out. Operating principle: When one of the signals S1 to Sn is logic 1, the M1 transistor is turned on, the node Vx is also zero voltage, M2 can be turned on, and the phase of the output signal can be selected. This architecture uses a smaller number of transistors, thus allowing for high-frequency operation. With careful design, the same delay time as two inverters in series can be achieved. Comparison of the linearity of the delay line of the two architectures is shown in Figure 4b. The simulation results show that the two delay line architectures can be designed with the same linearity. DNL analysis results show that the absolute value is less than 0.06, as shown in Figure 4c. As shown in Figure 4c, we can prove that the delay line of two inverter strings and the delay line of one inverter plus another inverter with selective signals (S1 to Sn) can achieve the same delay time.

### 4.2. Fine Detector and Fine Delay Line

Figure 5 shows the Vernier TDC architecture for fine-tuning detection. Two delay lines DL1 and DL2 are used. The delay time of DL1 using the delay cell is $\tau$A, and the delay time of DL2 using the delay cell is $\tau$B. The resolution of the fine-tuning circuit is $\tau$A-$\tau$B, and the phase difference T_Fine between Clk_Out and Clk_in is quantified, as shown in Figure 2. The arbiter circuit is used to compare the phase lead or lag. When the phase of DL1 leads the phase of DL2, the arbiter output is logic 1. When the phase of DL2 leads the phase of DL1, the arbiter output is logic 0, the quantized phase difference T_Fine is converted into a digital code Fine_code [1: $n$], and the generated Fine_code [1: $n$] controls

the fine delay line (FDL), as shown in Figure 2. The delay time of the H_Clk signal will be half of the F_Clk signal, so the generated Fine_code [1: $n$] only uses even bits.

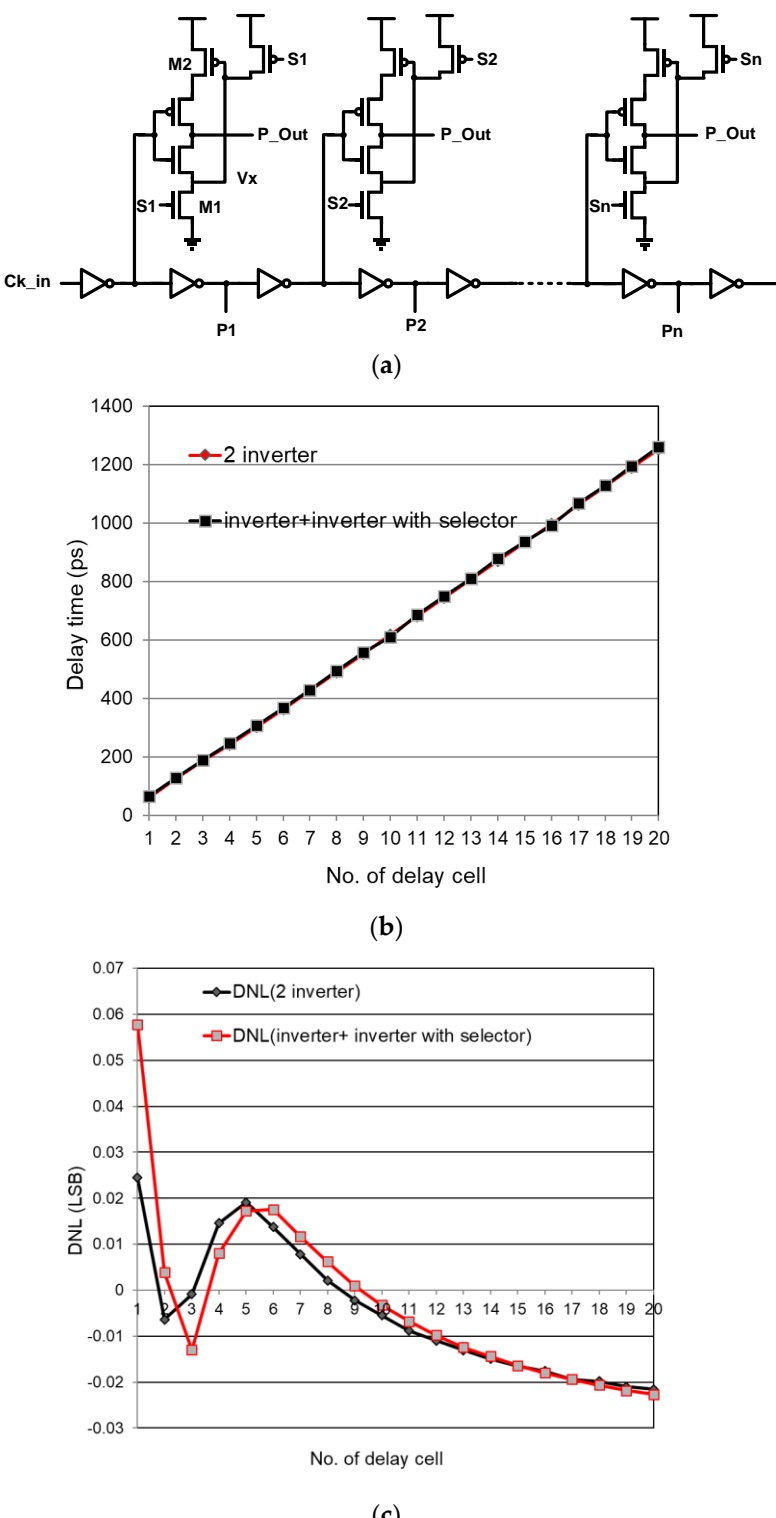

**Figure 4.** (**a**). Half Delay Line circuit; (**b**) Comparison of the linearity of the delay line of the two architectures; (**c**) DNL analysis results.

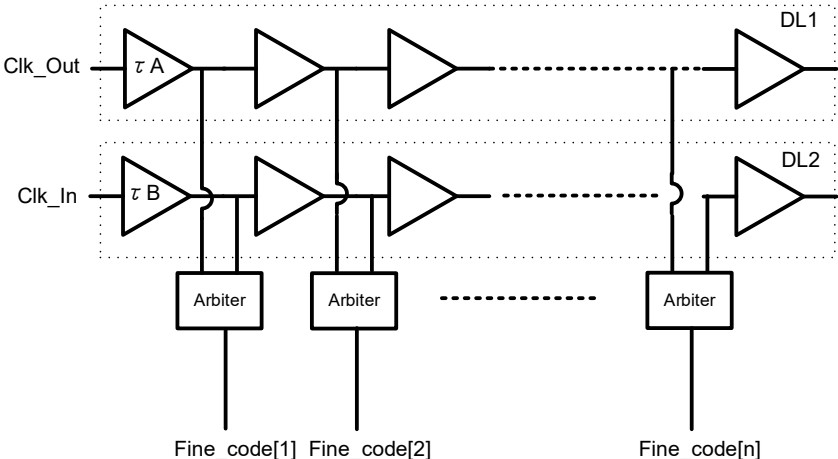

**Figure 5.** Fine detector.

Figure 6 shows a fine delay line (FDL) using two inverters connected in series, with 12 transmission gates added in the middle as a load. The fine-tuning digital code Fine_code [1: $n$] is used to change the delay time for the entire fine-tuning delay line. The delay time between each digital code is about 8.5 ps, and the delay time added by the design of the fine delay line is close to the time difference of $\tau A$-$\tau B$. Figure 7a shows the relationship between the delay time of the Fine Delay Line architecture and the input digital code Fine code [1: $n$]. The circuit of Figure 6 is simulated with the HSPICE tool to obtain the delay result, as shown in Figure 7a. Figure 7a clearly shows the linearity of the transmission gate. Figure 7b shows the capacitance of the transistor in Figure 6 can produce different capacitance values in different operating intervals. When the transmission gate is used as a load, and when the digital code is logic zero, the transistor of the transmission gate is operated in the Off region. The capacitance value and $WC_{OV}$ are related, as shown in Figure 7b, and when the digital code is logic 1, the transistor of the transmission gate operates in the triode region, as shown in Figure 7b. The source and drain are equipotential, so the capacitance is related to $WLC_{OX}/2 + WC_{OV}$. This architecture uses the capacitance of the transistor as the load capacitance. Compared with the previous use of capacitance and resistance as the fine-tuning delay unit, when the capacitance of the transistor is used as the load capacitance, it can produce a smaller delay time and a higher linearity, as shown in Figure 7a.

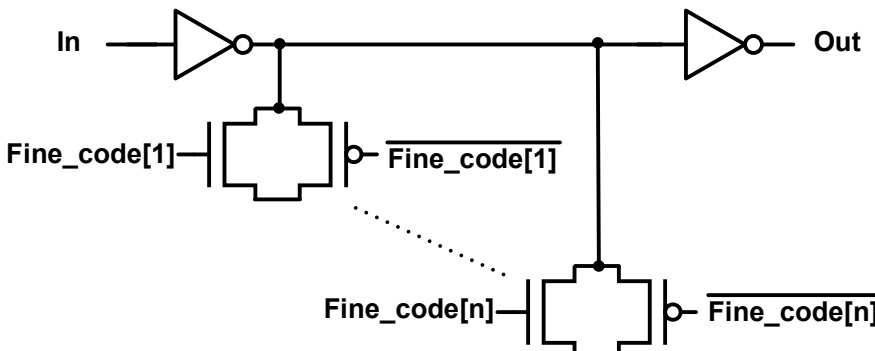

**Figure 6.** Fine Delay Line.

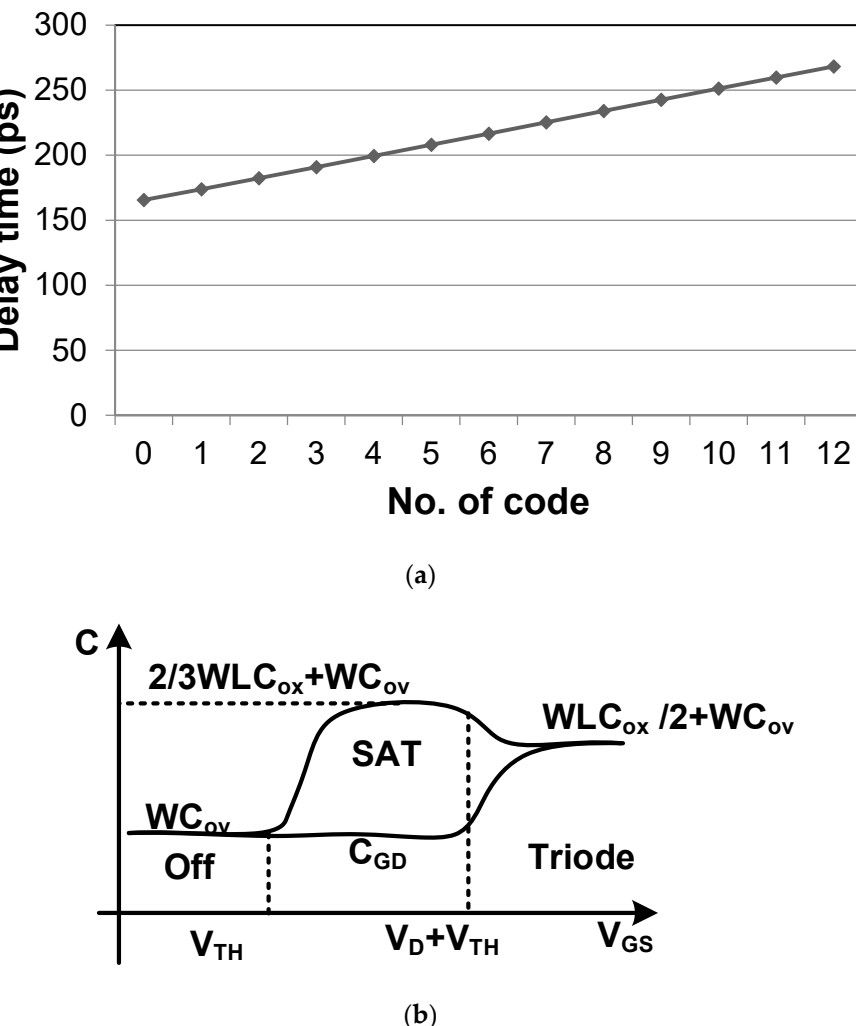

**Figure 7.** (**a**) Delay time of the Fine Delay Line; (**b**) Variation of capacitance versus $V_{GS}$.

## 5. Experimental Results

The manufacturing process used in this circuit is 0.18 μm, Figure 8 shows the chip micrograph, and the chip core area is 0.0613 mm². The input signal range of this circuit is 500 MHz to 1 GHz, and the duty cycle of the input signal ranges from 20% to 80%. Figure 9a,b show the 50% duty cycle of the output signal, when the input signal is 1 GHz and the duty cycle is, respectively, 20% and 80%. When the input signal is a 20% duty cycle, the output phase error is 24.9 ps and the duty cycle is 51.9%. When the input signal is a 80% duty cycle, the output phase error is 24.8 ps and the duty cycle is 51.6%. Given that the frequency of the input signal is 1 GHz and the power supply voltage is 1.8 V, the phase jitter value of the measured output signal is shown in Figure 10. The measured output phase jitter peak-to-peak value is 9.89 ps.

Figure 11 shows the error statistics for measuring the output signal for a 50% duty cycle. When the input frequency and period are different, the frequency of the input signal ranges from 500 MHz to 1000 MHz every 100 MHz, and the duty cycles of the input signal are 20%, 50% and 80%. Figure 11 shows that the duty cycle error is less than 2.5%. Figure 12 shows the measurement of the phase error statistics of the input and output signals. The measured results show that the phase error from 500 MHz to 1000 MHz is less than 25 ps. Figure 13 shows the measurement statistics of Peak-to-Peak Jitter and RMS jitter. When the power supply voltage is 1.8 V, the measured output peak-to-peak jitter is less than 13.04 ps, and the RMS jitter is less than 1.3 ps.

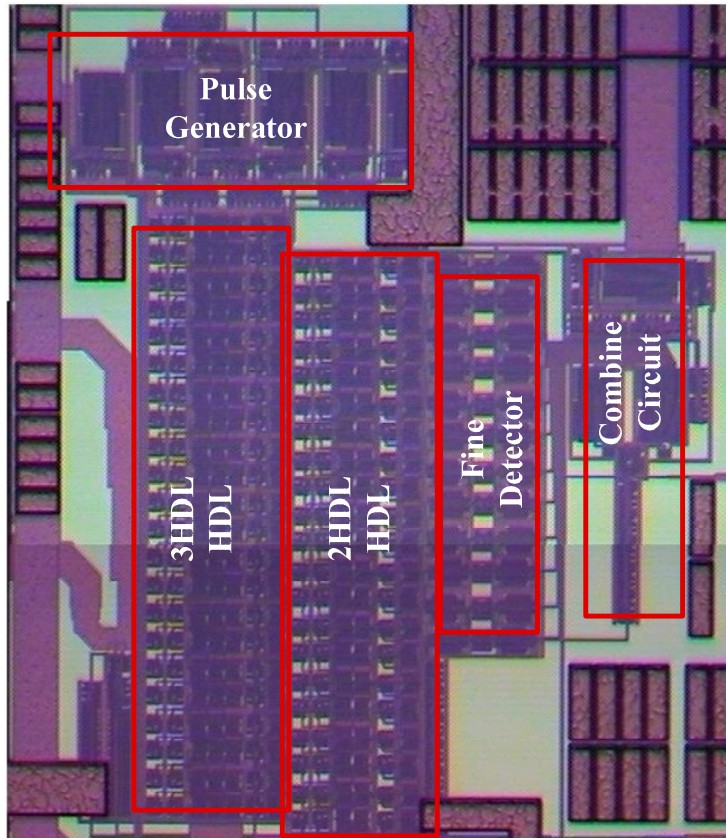

**Figure 8.** Chip micrograph.

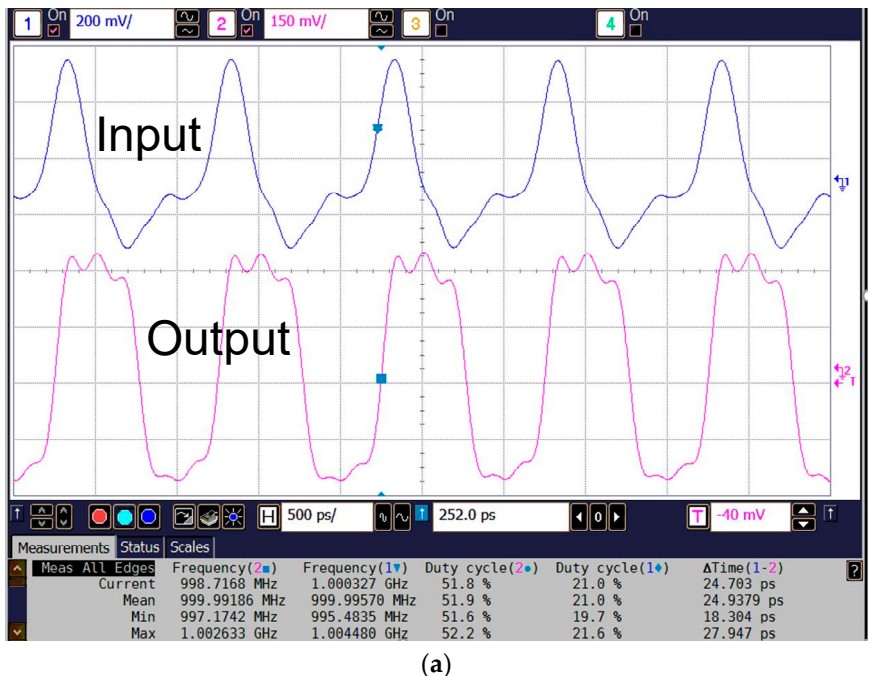

(**a**)

**Figure 9.** *Cont.*

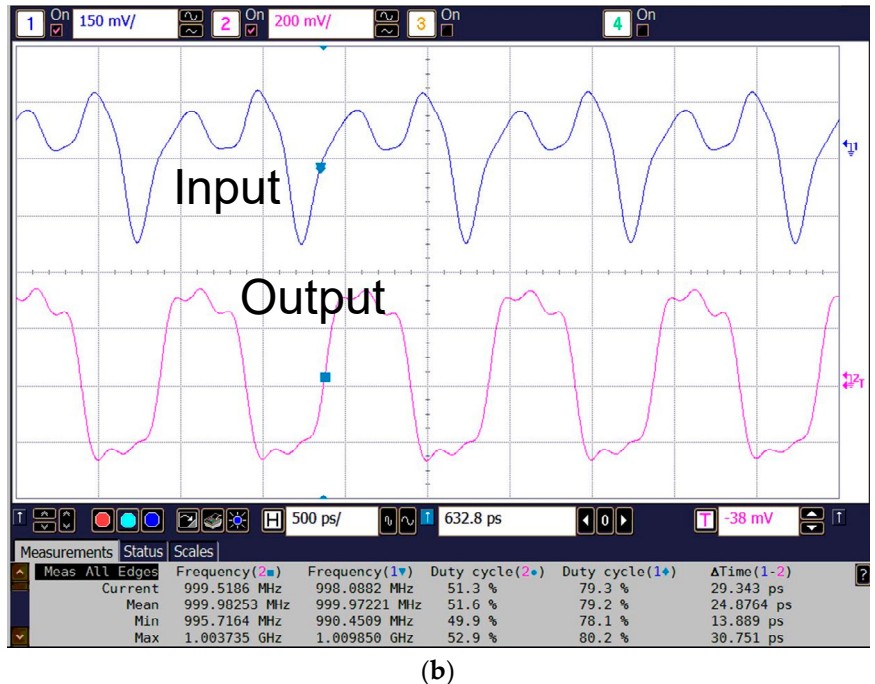

(**b**)

**Figure 9.** Measured result with input duty at 1 GHz (**a**) 20% (**b**) 80%.

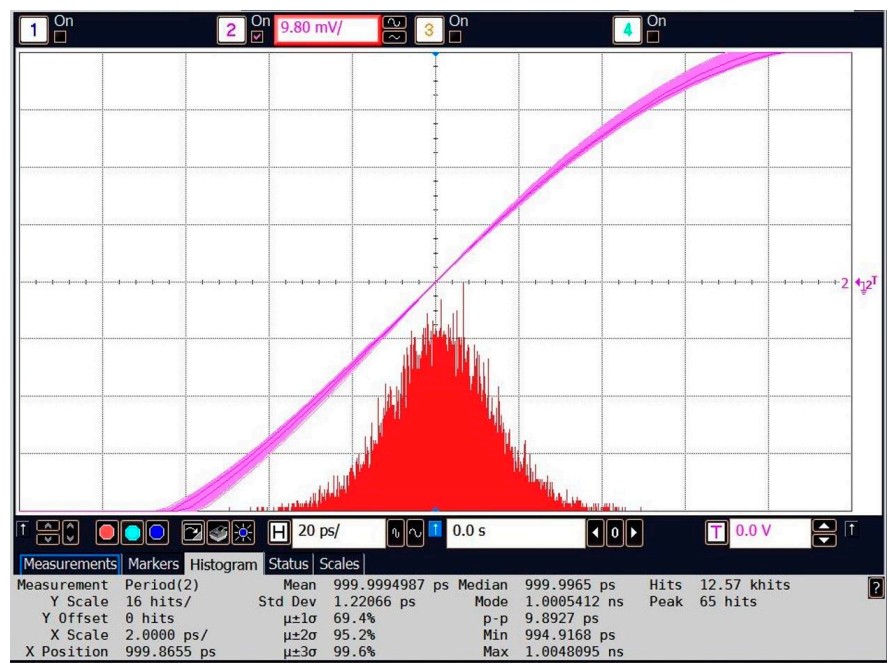

**Figure 10.** Measured jitter histogram at 1 GHz.

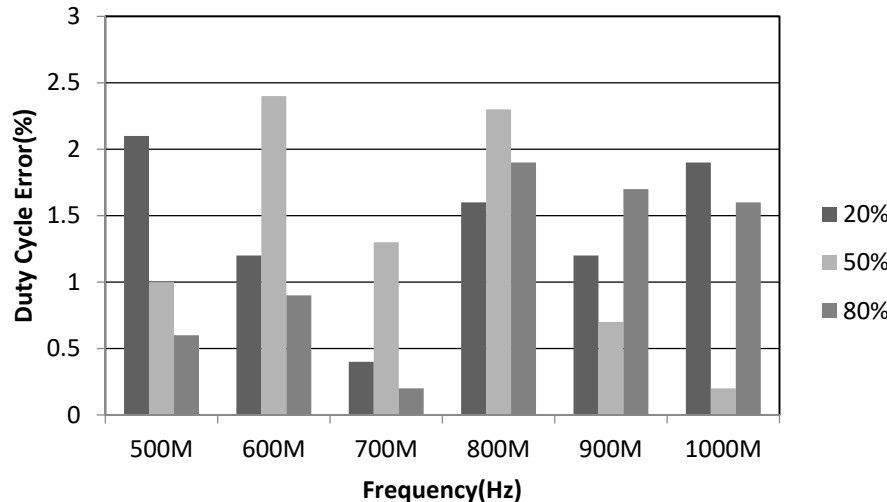

**Figure 11.** Duty cycle error with different input frequency.

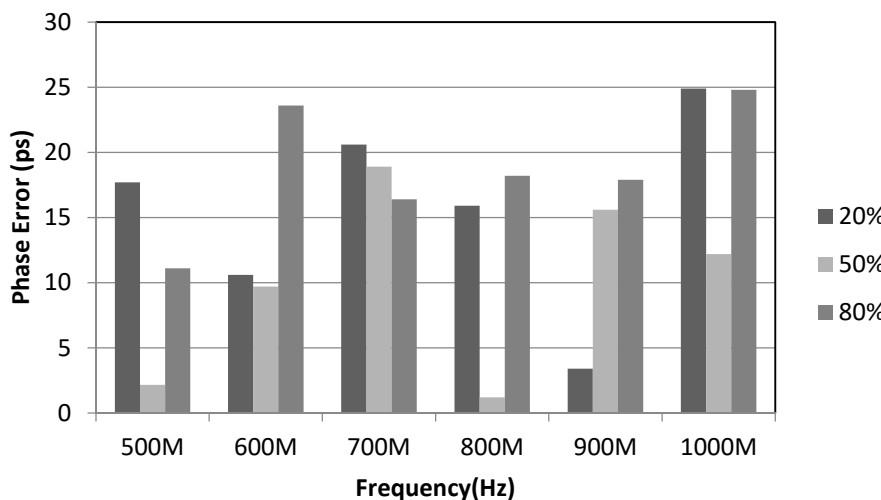

**Figure 12.** Phase error with different input frequency.

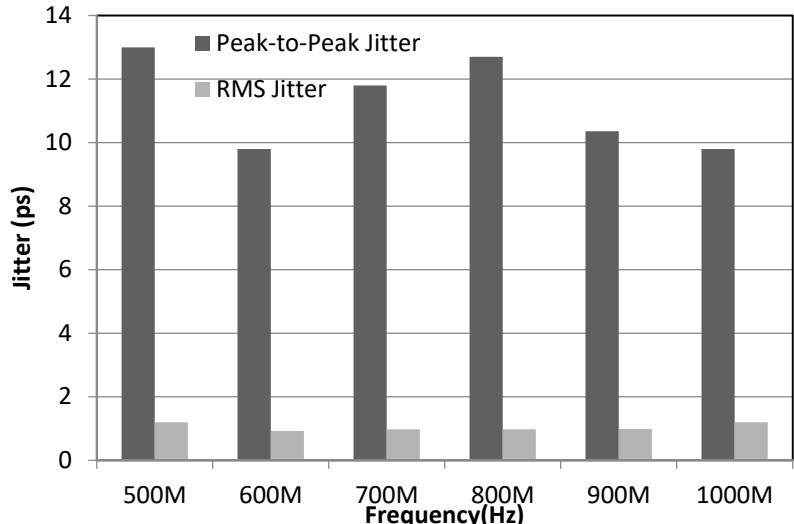

**Figure 13.** Peak-to-Peak Jitter and RMS jitter with different input frequency.

Table 1 compares the relevant specifications. Equation (4) is the FOM of this paper. The proposed circuits are compared against those from other studies. The comparison items include power, area lock time (Area), operating frequency range (Δfrequency), input duty cycle range (Δduty cycle), used process (Process), maximum frequency ($f_{max}$), locking cycle and operating voltage (VDD).

$$\text{FOM} = \frac{\text{Power} \times \text{Area} \times \text{Locking cycle}}{\Delta\text{frequency} \times \Delta\text{duty cycle} \times \text{Process} \times f_{max} \times VDD} \tag{4}$$

**Table 1.** Performance summary and comparison.

|  | [16] | [17] | [18] | [19] | [20] | [21] | [22] | This Work |
|---|---|---|---|---|---|---|---|---|
| Process (μm) | 0.065 | 0.090 | 0.13 | 0.18 | 0.18 | 0.055 | 0.055 | 0.18 |
| Supply voltage (V) | 1 | 1 | 1.2 | 1.8 | 1.8 | 1 | 1 | 1.8 |
| Correction Range (%) | 14–86 | 9–86 | 10–90 | 30–70 | 20–80 | 20–80 | 38–54 | 20–80 |
| Frequency Range (GHz) | 0.262–1.02 | 0.075–0.734 | 0.350–1 | 0.250–0.625 | 0.440–1.5 | 0.333–1 | 1–3 | 0.5–1 |
| Locking Time (cycles) | 24 | <15 | 14 | <36 | 15 | <5 | <275 | <6 |
| Duty-Cycle Error (%) | 1.4 | 1.78 | <1.4 | <1.6 | ±1.8 | 2 | 0.8 | <1.9 |
| Phase error (ps) | 484 | 688.4 | NA | NA | −10.6 | NA | NA | 24.8 |
| Active Area (mm$^2$) | 0.01 | 0.0289 | 0.059 | 0.09 | 0.053 | 0.016 | 0.003 | 0.0613 |
| Power Consumption (mW) | 6.5 | 4.59 | 5.6 | 10.8 | 43 | 2.09 | 2.08 | 10.1 |
| FOM | 0.431 | 0.594 | 0.57 | 11.52 | 1.11 | 0.09 | 0.27 | 0.382 |

The comparison Table 1 shows that the proposed circuit has a shorter lock-in period and in-phase with input clock. If a more advanced manufacturing process is used, the power consumption and circuit area can be further reduced.

## 6. Conclusions

This paper proposes an all-digital duty cycle correction circuit with fast locking, and adopts a new quantization method to effectively solve the problem that when the traditional time-to-digital converter produces a 180 degree delay phase or half delay of the input clock. The quantization error generated due to the odd number of digital codes being reduced to half the original value results in the loss of the original accuracy.

The proposed circuit uses a 0.18 μm process, the circuit operating frequency is 500 MHz to 1 GHz, and the acceptable wide range of input duty cycle is 20% to 80%. The proposed circuit can achieve synchronization and generate a 50% duty cycle output signal within 6 input signal cycle times. The measurement result shows that the phase error of the input and output signal is less than 25 ps, resulting in a 50% duty cycle error of less than 2.5%.

**Funding:** This work was supported in part by the Ministry of Science and Technology (MOST), Taiwan, under Grant MOST 107-2221-E-185-061.

**Acknowledgments:** The authors would like to thank the EDA tool supports and the chip manufacturer of Taiwan Semiconductor Research Institute (TSRI).

**Conflicts of Interest:** The authors declare no conflict of interest.

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
