# Peer review of "A 6-Locking Cycles All-Digital Duty Cycle Corrector with Synchronous Input Clock"

_electronics, doi:10.3390/electronics10070860_

Round 1

Reviewer 1 Report

 The paper presents a 6-locking Cycles All-Digital Duty Cycle Corrector with Synchronous Input Clock with fast locking implemented in 0.18um CMOS process.

 Introduction is well written and presents a good state of the art approach.

 In the second section it is presented the circuit architecture design that uses two adjacent rising edges of the input clock to be input to two delay lines with different delay times. This solution is well presented in this work.

 Remove the : in line 119.

 Figure 3 shows in a clear form the proposed method.

 How is obtained  Figure 7 ? Is supported by any expression or it's given by the manufacturer.

 Section 4 presents a very carefully and detailed presentation of the experimental results.

 Conclusions are well written and perceptible with the FOM used to compare with other works.

Reviewer 2 Report

The article addresses a topical issue in the field of integrated circuits used at high frequencies and suggests a solution for Duty Cycle Corrector that offers high accuracy according to tests.
It would be interesting to examine a comparison with the solutions currently used in mass production of circuits. Also, an appreciation of the complexity of this correction system and of the chances of it being implemented on a large scale would be extremely useful.

Reviewer 3 Report

The duty cycle corrector is for controlling circuit triggering under high-frequency operation; the accuracy of the clock signal's frequency, phase, and pulse width is essential for digital systems.

Recommendations
The introduction is a bit large. I would split the section "introduction" into two subsections: "Introduction"  and "State-of-the-art", first from line 17-36, second from 37-115. After the introduction, I would write what comes next, like: "Section 2 presents the State-of-the-art", in section 3, there is the explanation of the Circuit Architecture Design, etc.

There is a thorough explanation of the digital system and comparison with the state-of-the-art, table 1. The author explained the correctness of the system satisfactorily, and the manuscript can benefit other digital designers.
